
# Delineation of discrete conduit networks in karst aquifers via combined analysis of tracer tests and geophysical data

Jacques Bodin[1], Gilles Porel[1], Benoît Nauleau[1], Denis Paquet[1]

[1]Université de Poitiers, CNRS, UMR 7285 IC2MP, 40 Avenue du Recteur Pineau, 86022 Poitiers Cedex, France

*Correspondence to*: Jacques Bodin (jacques.bodin@univ-poitiers.fr)

**Abstract.** Assessment of the karst network geometry based on field data is an important challenge in the accurate modeling of karst aquifers. In this study, we propose an integrated approach for the identification of effective three-dimensional (3D) discrete karst conduit networks conditioned on tracer tests and geophysical data. The procedure is threefold: i) tracer breakthrough curves (BTCs) are processed via a regularized inversion procedure to determine the minimum number of distinct

tracer flow paths between injection and monitoring points, ii) available surface-based geophysical data and borehole-logging measurements are aggregated into a 3D proxy model of aquifer hydraulic properties, and iii) single or multiple tracer flow paths are identified through the application of an alternative shortest path (SP) algorithm to the 3D proxy model. The capability of the proposed approach to adequately capture the geometrical structure of actual karst conduit systems mainly depends on the sensitivity of geophysical signals to karst features, whereas the relative completeness of the identified conduit network

depends on the number and spatial configuration of tracer tests. The applicability of the proposed approach is illustrated through a case study at the Hydrogeological Experimental Site (HES) in Poitiers, France.

## 1 Introduction

Karst conduits in carbonate aquifers provide low-hydraulic resistance paths, which exert a strong control on the spatiotemporal propagation of pressure-head perturbations (e.g., pumping-induced drawdowns) and the transport of solutes in groundwater

(Goldscheider and Drew, 2007; Worthington and Ford, 2009; Kresic, 2012; Ronayne, 2013). It is therefore advisable that karst conduit networks are explicitly represented in numerical hydrogeological models (Worthington, 2009; Saller et al., 2013; Malard et al., 2015). However, due to the relative inaccessibility of aquifers, accurate characterization of the karst network geometry is a challenging task. Even in cases where portions of the karst system can be explored and mapped by speleologists and/or cave divers (Gallegos et al., 2013; Lauber et al., 2014; Scharping et al., 2018; Vuilleumier et al., 2019), it is widely

acknowledged that known and accessible karst conduits represent only a small fraction of the whole drainage network. In most karst aquifer studies, the spatial occurrence of karst conduits is only known at sparse locations corresponding to their intersection with the ground surface (sinkholes and springs) and/or with boreholes.

A number of approaches have been proposed to delineate karst networks according to their observed location. Among the methods is the generation of plausible conduit networks either through analog templates (Pardo-Igúzquiza et al., 2012;





Fournillon et al., 2012; Le Coz et al., 2017) or via the simulation (mimicking) of the action of speleogenetic processes considering pre-existing rock discontinuities (fractures, bedding planes, and inception horizons), e.g., Jaquet et al. (2004), Borghi et al. (2012) and De Rooij and Graham (2017). Other approaches strive to infer the spatial distribution of karst conduits through the inversion of multiple pumping test and/or tracer test data, also referred to as hydraulic or tracer tomography (Borghi et al., 2016; Mohammadi and Illman, 2019; Fischer et al., 2020). Regardless of the method pursued, any data describing the

likely occurrence and location of karst conduits in the subsurface should be supplied to a given model. Geophysical methods are appealing for this purpose because of their potential to image the subsurface in a noninvasive and quasi-continuous manner. However, the detection and mapping of karst conduits based on geophysical surveys remains a challenging task. As pointed out by Chalikakis et al. (2011), the amplitude of the geophysical anomalies associated with karst conduits highly depends on their size and depth. The resolution of geophysical imaging techniques also depends on the spatial density of field

measurements, which is inherently limited for practical reasons. To date, reported field applications of surface geophysical methods to locate known or suspected karst conduits in the subsurface have only been successful in regard to large-diameter (>1 m) conduits at shallow depths (<20 m), e.g., Guérin et al. (2009), Zhu et al. (2011) and Sawyer et al. (2015). Relying on geophysical surveys alone for the delineation of karst networks, applicable to very small and deep karst conduits, is therefore hardly conceivable. More generally, it has been increasingly agreed that geophysical imaging methods are not "silver bullets"

(Singha, 2017) and that geophysical data should be analyzed conjointly with hydrogeological data for a more accurate characterization of flow- and transport-relevant heterogeneities (Hyndman and Gorelick, 1996; Rubin and Hubbard, 2005).

In the present study, we investigate the delineation of karst conduit networks via the joint analysis of tracer test data, three-dimensional (3D) seismic images, and borehole flow measurements. To the authors' knowledge, the only previous study to map karst conduit networks based on geophysical data is that of Vuilleumier et al. (2013). In this study, two-dimensional (2D)

airborne electromagnetic data were processed with the pseudogenetic karst simulator of Borghi et al. (2012). At the core of the method is the computation of the shortest paths (SPs) (minimum effort) in the heterogeneous medium depicted by geophysical surveys with a fast marching algorithm (Sethian, 1996). The above SP search is also central to the approach developed in the present study. However, our approach differs from the work of Vuilleumier et al. (2013) in three main aspects. First, instead of processing 2D electromagnetic data, we process 3D seismic data supplemented by borehole flow measurements. Second,

rather than applying a single path routing algorithm in the delineation of karst conduits between pairs of source-target locations, we adopt a multiple (alternative) path finding algorithm. The applied routing scheme allows for the mapping of diverging-converging paths, which are common in karst systems, e.g., Collon et al. (2017) and Jouves et al. (2017). Finally, interwell tracer test data are considered to maximize the information value of geophysical data. More precisely, tracer breakthrough curves (BTCs) are inverted with a recently developed software (Bodin, 2020) prior to the processing of geophysical data. BTC

inversion allows us to identify the minimum number of flow paths between the tracer injection and monitoring locations, which is subsequently applied in the multiple path routing formulation.

The paper is organized as follows. Section 2 develops the method for the determination of the minimum number of distinct flow paths involved in a tracer experiment. In section 3, we examine the use of geophysical data as a surrogate (proxy) for





aquifer hydraulic properties, and we present the multiple path finding algorithm applied to the proxy model. Section 4
illustrates the application of the proposed approach to map the conduit network within the karst aquifer at the Hydrogeological
Experimental Site (HES) in Poitiers, France. A discussion and conclusions are provided in section 5.

## 2 Assessing the minimum number of distinct flow paths involved in a tracer experiment

Artificial tracer testing is a widely adopted method for the characterization of karst aquifers. Experiments are typically
conducted by injecting a tracer-labeled solution into a sinkhole or well and subsequently monitoring the tracer concentration
response at one or several downstream locations, usually spring(s) or pumping well(s). In the following, we will restrict the
discussion to the case of tracer tests performed using nonreactive tracer species under steady-flow conditions and with a much
shorter duration of the injection signal than the mean tracer transit time. Given the possibly complex conduit network patterns
in karst aquifers, the tracer may follow different routes between the injection and monitoring points. As a result, tracer BTCs
often exhibit multiple local peaks and/or extensive backward tailing, e.g., Kübeck et al. (2013), Labat and Mangin (2015) and
Barberá et al. (2018).

The determination of the actual number of transport flow paths involved in a tracer experiment is a challenging task because
a given flow system containing $N$ transport flow paths may produce a BTC exhibiting 1 to $N$ concentration peaks. The number
of distinguishable peaks depends on three factors: (i) the difference between the mean travel times (advection), (ii) the variance
in the travel time distribution (dispersion) along each flow path, and (iii) the relative exchange (mixing) between the flow
paths. On the other hand, unimodal heavy-tailed BTCs do not necessarily indicate the occurrence of multiple overlapping
pathway responses. The long-tail behavior of a BTC may also indicate solute mass exchange between a single flow pathway
and adjacent stagnant water zones, which may reflect various features: primary rock porosity, dissolution vugs, pool volumes,
fragmented rock areas, transverse dead-end conduits or fractures, etc.

We propose to assess the number of distinct tracer flow paths via an inverse modeling procedure implemented in MFIT
software (Bodin, 2020). MFIT is a BTC fitting tool that combines different analytical transport models with PEST optimization
routines (Doherty, 2019a, 2019b). The general modeling approach is based on the multiflow framework first introduced by
Maloszewski et al. (1992), which assumes that the karst network structure between the injection and monitoring points can be
approximated by a combination of independent one-dimensional channels (Fig. 1). The four analytical models implemented
in MFIT to describe the transport process at the scale of individual channels include (i) the solution of the classical advection-
dispersion equation (ADE) for an instantaneous point source (Kreft and Zuber, 1978), (ii) the solution of the ADE regarding
an exponentially decaying injection pulse (Marino, 1974), (iii) the single-fracture dispersion model (SFDM) of Maloszewski
and Zuber (1990), and (iv) the two-region nonequilibrium (2RNE) model of Toride et al. (1993). Both the SFDM and 2RNE
models are double-porosity models that consider solute mass exchange between channels and adjacent stagnant water zones.
A fundamental difference between these two models is the mathematical description of the exchange between the mobile and
immobile regions, which is assumed to be governed by a first-order process in the 2RNE model and by a second-order





(diffusion) process in the SFDM. A complete mathematical description of these models has been provided by Bodin (2020) and is not repeated here for conciseness. While it is acknowledged that the actual geometry of the karst conduit system experienced by a tracer could be much more complex than that depicted in Fig. 1, it is assumed that this approach allows us to capture the effects of tracer mass routing through distinct pathways. In other words, channels are not assumed to represent

individual karst conduits but are rather regarded as lumped submodels of the main flow routes through the karst network.

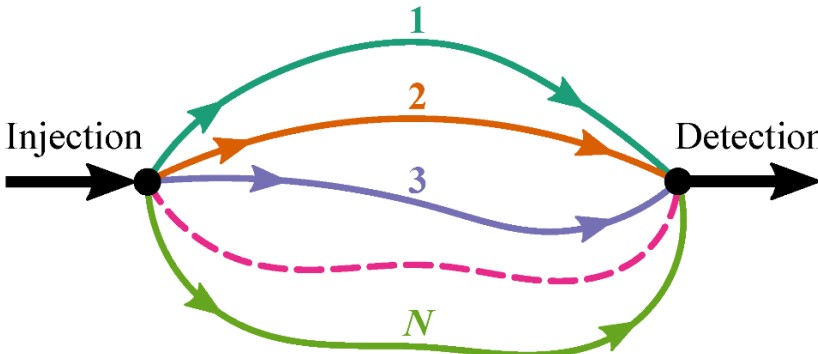

**Figure 1.** Schematic layout of the multiflow modeling approach implemented in MFIT software (Bodin, 2020). The tracer transport from the injection site to the monitoring point is assumed to occur in a flow network comprising $N$ independent one-dimensional channels.

The fitting process of a model curve to an experimental tracer BTC first requires specification of the number of channels $N$, followed by optimization of the model parameters pertaining to each channel. It is easily conceivable that increasing $N$ allows for a finer adjustment of the experimental BTC but also increases the risk of overfitting. In MFIT, the fitting error objective function is referred to as PHI and is computed as the sum of the squared weighted differences between the tracer BTC and the model-simulated curve. Determination of the optimal (minimum) number of channels, hereafter referred to as $N^*$, is achieved

through analysis of the Pareto curve, which represents the minimum PHI value obtained at different $N$ values. The typical shape of a PHI($N$) Pareto curve is that of a monotonic decreasing function (please refer to Fig. 2). The optimal $N^*$ value corresponds to the inflection point on the PHI($N$) curve, i.e., where an increase in the number of channels does not substantially improve the model fit. The $N^*$ value is also a model-dependent parameter, as fewer channels are required to fit a heavy-tailed BTC with a double-porosity model (SFDM or 2RNE) rather than with the ADE instantaneous injection model. As a side note,

care must be taken to prevent numerical instability in the optimization procedure, which may create artifacts in the computed PHI($N$) curves and therefore compromise the determination of $N^*$. These problems may arise from transport model nonlinearity and the possibly large number of fitting parameters. The combined use of the two regularization methods implemented in PEST, namely, truncated singular value decomposition and Tikhonov regularization, ensures the stability of the inversion process.

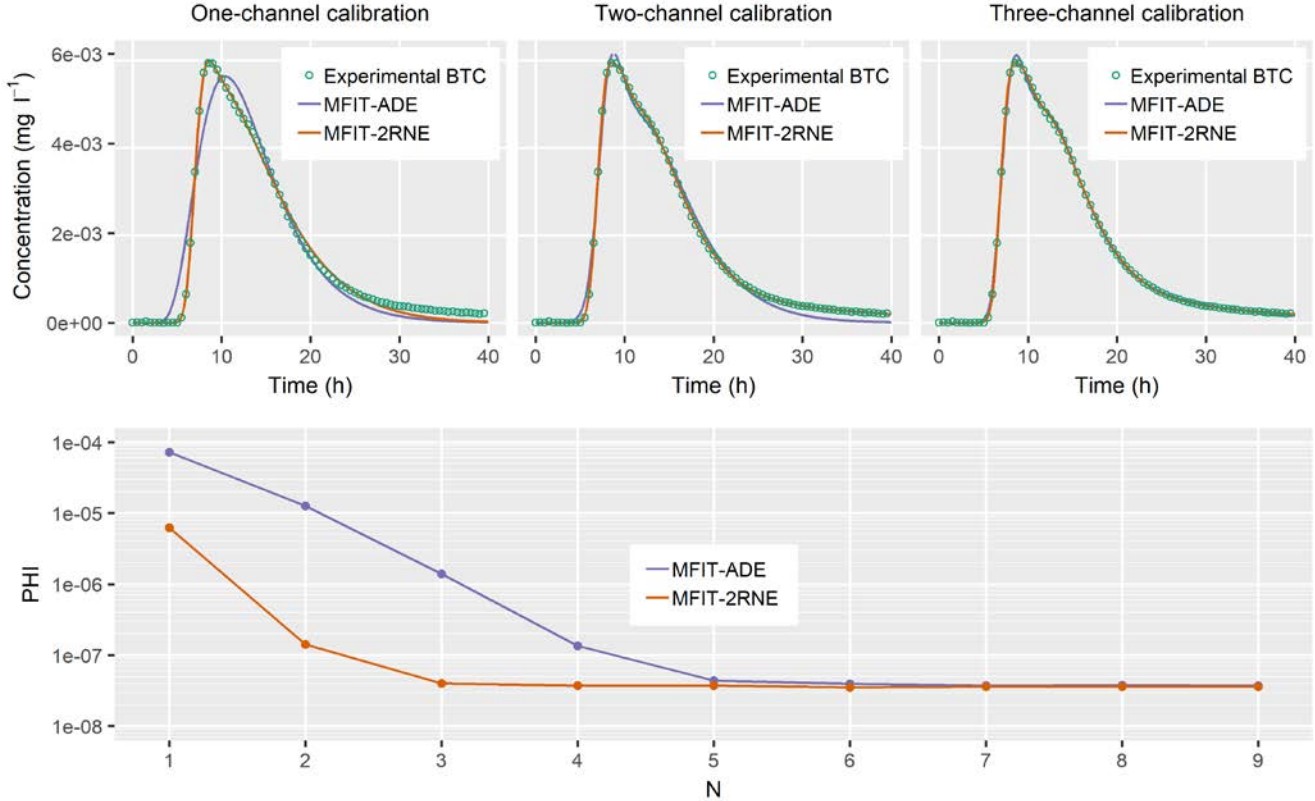

**Figure 2.** Example of BTC fitting analysis with the multiflow ADE and 2RNE models for different numbers of channels N. The experimental BTC corresponds to a tracer test performed in 2016 at the HES between wells M16 (injection) and M22 (pumping and observation). PHI is the fitting error objective function (sum of the squared errors between the tracer BTC and model-simulated curve) minimized with the regularization routines in PEST. The decreasing trend of the PHI($N$) Pareto curves indicates the improvement in model fit with increasing number of channels $N$. The optimal numbers of channels determined with the ADE and 2RNE models are $N^* = 5$ and $N^* = 3$, respectively.

## 3 Proxy model of the aquifer hydraulic properties and multiple path finding

Considering that karst conduits provide paths with the lowest hydraulic resistance within a given aquifer, an $N^*$–conduit system between a pair of tracer injection and monitoring points can be identified by searching the $N^*$ optimal (most efficient) paths within the corresponding hydraulic conductivity field. The karst network can then be sequentially constructed by abutting/joining the subconduit systems identified based on different pairs of tracer injection and monitoring points. As our knowledge of the hydraulic conductivity field is often incomplete and uncertain, it is convenient to substitute the geophysical field for the hydraulic conductivity field in the search for optimal paths. A necessary assumption for this approach is that a





monotonic and spatially stationary relationship exists between the geophysical and hydraulic conductivity fields. However,
the relationship between both parameters does not need to be explicitly modeled, which is of particular interest given the
notoriously complex and site-specific nature of the problem, e.g., Pride (2006), Hyndman and Tronicke (2006) and Brauchler
et al. (2012). It is beyond the scope of this paper to examine the potential strengths and weaknesses of the different existing
geophysical methods for the characterization of hydrogeological heterogeneity. We refer the reader to the comprehensive
review conducted by Binley et al. (2015). Of course, the reliability of the conduit networks identified with our approach largely
depends on the sensitivity of geophysical measurements to karst features. It is also possible to combine different types of
geophysical data into a single proxy model of aquifer hydraulic properties. In the application example presented in section 4,
we adopted both seismic imaging and borehole flow logging techniques.

Once a proxy model of the aquifer hydraulic properties has been established, the main challenge is the identification of the
optimal paths with this model. The determination of the most efficient path between two points is a classical problem in graph
theory and is commonly referred to as the SP problem. Basically, a graph is specified by a set of vertices (nodes) and a set of
edges (links) connecting the vertices. Each edge is assigned a weight (cost) of traversing. This abstraction can be applied to
any gridded model by mapping the vertices onto the centers of model grid cells and by connecting each vertex to its neighbors
via a regular edge network. Building on the assumed (positive or negative) relationship between the geophysical and hydraulic
conductivity fields, pseudolocal hydraulic resistance coefficients can be assigned to the edges as the mean (or inverse mean)
geophysical property value of the two connected grid cells. A number of SP algorithms have been developed since the early
1960s, e.g., the reviews by Cherkassky et al. (1996) and Fu et al. (2006) and the recent works of Song et al. (2018) and Arslan
and Manguoglu (2019). A classical and widely adopted algorithm is that of Dijkstra (1959), which was used, e.g., by Knudby
and Carrera (2006) and more recently by Rizzo and de Barros (2017), in the determination of the path of least hydraulic
resistance in synthetic hydraulic conductivity fields. Borghi et al. (2012) applied the fast marching algorithm of Sethian (1996)
to simulate karst conduit networks as least-resistance paths based on 3D scalar grids of pseudovelocity fields empirically
derived from different geological indicators. The fast marching method can be regarded as an extension of Dijkstra's method
to the continuous domain. In a study parallel to that of Borghi et al. (2012), Collon-Drouaillet et al. (2012) adopted the A*
algorithm (Hart et al., 1968), which utilizes a heuristic function to guide the search process. For our purpose, we are interested
not only in determining the SP between two vertices but also in determining a finite number ($N^*$) of distinct paths, as revealed
by tracer BTC inversion. In graph theory, this problem is known as the k-shortest path (KSP) problem, and many algorithms
have been proposed to solve this problem, e.g., Yen (1971), Lawler (1972), Brander and Sinclair (1996), Eppstein (1998),
Hershberger et al. (2007), Scano et al. (2015) and Chondrogiannis et al. (2015, 2017) and the references therein. In the present
study, we selected the OnePass+ algorithm of Chondrogiannis et al. (2015, 2017) as implemented in the Alternative Routing
Library for Boost Graph (ARLib) developed by Leonardo Arcari (https://github.com/leonardoarcari/arlib, last accessed: 10
November 2021). The main interest of this algorithm is that it allows the user to specify a maximum overlap ratio between
alternative paths, according to the following expression:





$$\frac{\sum_{(n_x,n_y)\in p\cap p'} w_{xy}}{\sum_{\forall (n_x,n_y)\in p'} w_{xy}} \leq \theta \tag{1}$$

where $p$ and $p'$ denote any pair of alternative paths from the KSP solution set, $w_{xy}$ is the weight of an edge $(n_x, n_y)$ belonging to $p$ and/or $p'$ with the exception of the starting and ending edges, $p\cap p'$ denotes the set of edges shared by $p$ and $p'$, and $\theta\in[0,1]$ is the similarity threshold. With this threshold, the OnePass+ algorithm allows the identification of alternative paths that are significantly dissimilar from each other. This is particularly relevant for dense graphs retrieved from a continuous geophysical model. Otherwise, the KSP search process typically results in a set of very similar paths with only minor deviations with respect to the SP. An open issue is the specification of the threshold value. As already noted, the $N^*$ paths yielded by the inversion of tracer test data are not supposed to be fully disconnected, as shown in Fig. 1, but their overlap must nevertheless be small enough to produce a detectable signature in the BTC. Further theoretical and/or numerical investigations could be pursued to narrow, if possible, the value range of $\theta$, but such developments are beyond the scope of the present study. Alternatively, $\theta$ may be considered a calibration parameter whose value may be adjusted to produce a good geometrical match between the computed conduit network and independent observed karst features. In the case study presented below, we adopted an empirical threshold of 0.5.

## 4 Case study: Hydrogeological Experimental Site (HES) in Poitiers, France

The investigations conducted at the HES focus on a confined limestone aquifer extending approximately 35 to 130 m below the ground surface. Detailed information on the hydrogeology of the site can be found in the article by (Audouin et al., 2008) and is not repeated here for the sake of conciseness. Only information considered relevant to the present study is included below. To date, the facility consists of 45 boreholes in an overall area of 15 ha, of which 28 boreholes are located within a square area of 210 m × 210 m (Fig. 3). Borehole camera surveys revealed the presence of karst conduits in the aquifer, with sections of up to 3 m². The karst features seem to preferentially occur in 3 specific lithostratigraphic units, which are subhorizontal (dip smaller than 2°), between 2 and 5 m thick, and located 50, 90, and 115 m below the ground surface. The bedded structure of these karst conduits is also supported by observations of rock outcrops located a few km from the HES in the same lithostratigraphic horizons (Fig. 4). It is difficult to state whether the karst conduits present in a given horizon are all interconnected or whether they form different disconnected clusters. Similarly, it is difficult to determine whether natural high-permeability connections occur between the different karst layers, for example, through vertical fractures. However, there is evidence that certain boreholes provide connections by intersecting karst conduits in the different horizons. Even in the absence of pumping, vertical flows can be measured in these boreholes at velocities of up to several meters per minute, thus indicating a natural difference in hydraulic head between the various karst horizons.





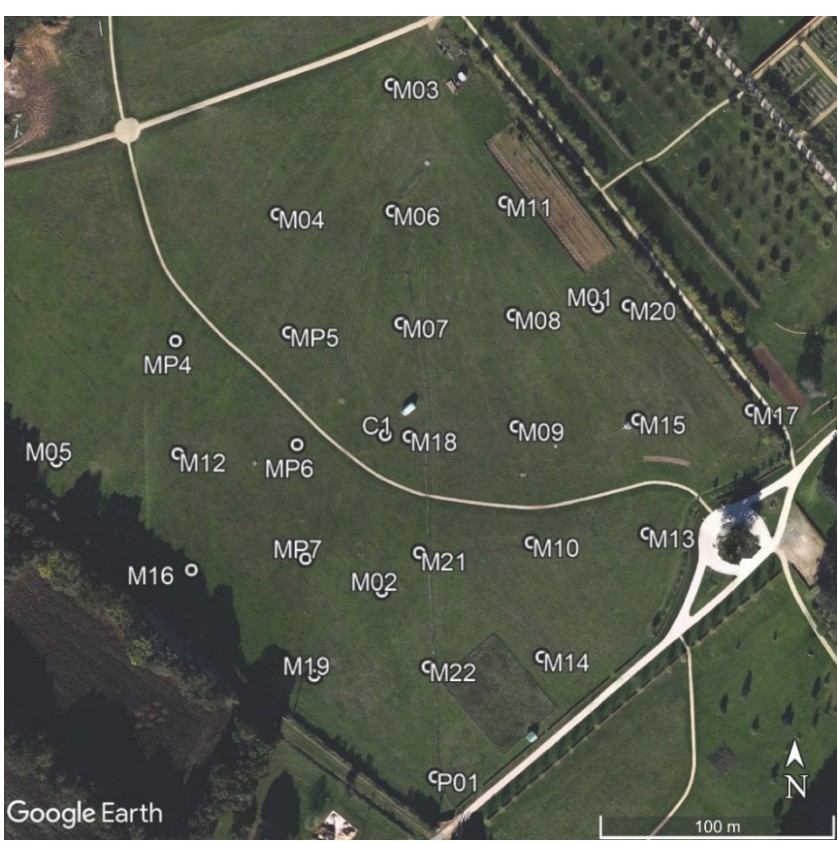

**Figure 3.** Location of the boreholes at the HES in Poitiers, France. Map data are retrieved from © Google Earth.

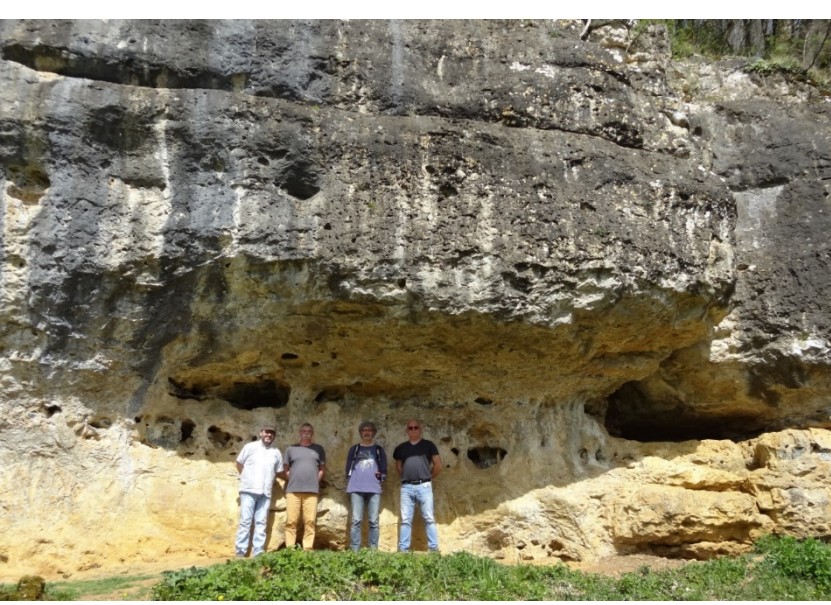

**Figure 4.** Bedded structure of karst conduits as observed on a rock outcrop located 5 km south of the HES.



A number of interwell tracer tests have been carried out at the HES over the last five years. The routine protocol applied in
these tests can be summarized as follows:

1) A pumping operation is initiated at a constant rate ranging from 50–65 m³/h, and a pseudosteady-state flow regime is
eventually established, i.e., stabilization of the hydraulic head gradient over the HES area. The typical time for this regime to
be reached is approximately 6 hours, but usually a minimum of 24 hours is observed before moving on to the next step.

2) Borehole flowmeter surveys are performed in the candidate wells for tracer injection: vertical flow velocities are measured
and possible inflow/outflow horizons are identified (Fig. 5).

3) The well and injection depth of the tracer are selected. The targeted injection depth is generally a few tens of cm upstream
of an outflow (from the well to the aquifer) horizon. For instance, Fig. 5 indicates that a suitable injection depth for a tracer
experiment between M02 and MP6 would be between 58 and 60 m depth.

4) Pipes 2.5 m in length and 1.5 cm in internal diameter are connected down to the targeted injection depth. This pipeline ends
with a screened cap that ensures horizontal diffusion of the tracer at the outlet.

5) The tracer solution (typically 5 g of uranine diluted in 2 L of water) is injected, followed by a water flush volume of 40 L.
The total injection duration, including flushing, is always less than 3 min.

6) The concentration at the outlet of the pumped well is monitored with a flow-through fluorometer (Albillia GGUN) connected
to a bypass of the discharge pipeline.

The BTC dataset adopted in the present study corresponds to 50 tracer tests conducted between wells located in the area
covered by seismic surveys. Four pumping wells were employed in the tracer experiments: M06, M07, M22, and MP6. Each
BTC was processed with MFIT using both the multiflow ADE and 2RNE transport models. The multiflow SFDM was not
adopted here because the typical duration of the considered tracer experiments, from a few hours to a few days, was insufficient
for diffusion processes to be of influence. Actually, more precisely, this model could be fitted to the obtained HES tracer BTCs
but without a better performance than that of the multiflow ADE model, which is simpler, or only if unrealistic diffusion
parameter values were considered, e.g., Bodin (2020). The optimal (minimum) number of channels $N^*$ required to fit the above
BTCs with the multiflow ADE and 2RNE models was determined as described in section 2. The results are given in Appendix
A, Table A1. The median $N^*$ values obtained with the multiflow ADE and 2RNE models are 5 and 3, respectively. Rather than
considering any form of competition between these two models, which are based on two different conceptual and mathematical
approaches to the description of tracer migration in a heterogeneous medium, we instead believe that the $N^*$ values associated
with the multiflow ADE and 2RNE models allow us to differentiate between primary and secondary karst paths. The primary
paths are those identified by the multiflow 2RNE model, while the secondary paths are the additional low-flow velocity
channels explicitly required by the multiflow ADE model to fit the curves, where the multiflow 2RNE model simulates the
exchange between the primary channels and surrounding stagnant water zones.





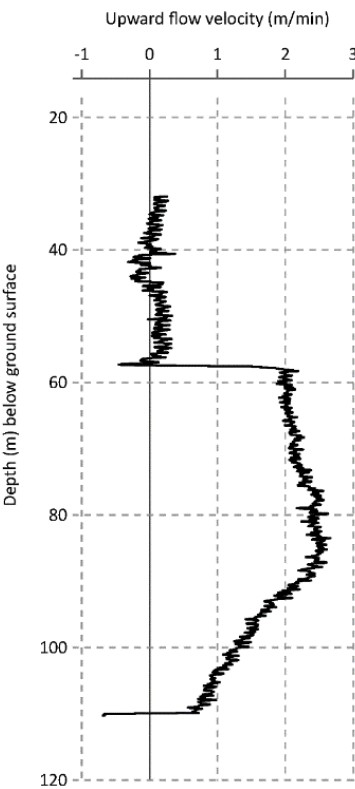

**Figure 5.** Example of borehole flowmeter data. The logged well is M02, while pumping is active in MP6 at a flow rate of 60 m³/h. The flow velocity measurements indicate an upward flow from approximately 110 to 55 m below the ground surface. In this same pumping experiment, downward flows are measured in other boreholes. In this case, the curve is to the left of the vertical axis (negative upward flow velocity values).

A 3D seismic survey was conducted at the HES in 2004. The data were acquired along 21 parallel profiles running SW-NE and equidistant by 15 m. The SW-NE direction corresponds to a locally known structural (i.e., fracturation and karst) direction and is that of a small topographic valley in which boreholes M03, M04, MP4 and M05 were drilled. Each profile consisted of 48 geophones with an interdistance of 5 m. Five dynamite shots (25 g per shot) were fired for each line, one shot at each end and three perpendicular shots at 40, 50, and 60 m from the center of the line. In addition to the surface seismic surveys, full

waveform acoustic logs were acquired in 5 boreholes (C01, MP5, MP6, M08, M09), and a vertical seismic profile (VSP) was acquired in C01. The procedure used to build the 3D seismic velocity model from the acquired data includes amplitude recovery, deconvolution, wave separation, normal move-out corrections and time versus depth conversion based on VSP measurements. The acoustic logs in wells C01, MP5, MP6, M08 and M09 were used as calibration constraints for the seismic depth model. The interested reader can find further details in the articles by Mari and Porel (2008), Mari et al. (2009), Mari et

al. (2020), and Mari and Porel (2021). The original seismic model grid consisted of 96 columns of 2.5 m in the SW−NE




direction, 60 rows of 5 m in the NW−SE direction, and 512 layers of 0.5 m in the vertical direction from 0 to 256 m below the ground surface. In the present study, we truncated this model vertically by retaining only the 190 layers corresponding to the extension of the aquifer from 35 to 130 m below the ground surface (Fig. 6). As discussed in the papers by Mari et al., the cross-analysis of the 3D seismic model and the well logs shows a close relationship between low-seismic velocity zones and

the main inflow/outflow horizons associated with high hydraulic conductivity features. The work presented below is based on the generalized assumption of a negative correlation between the seismic velocity and hydraulic conductivity.

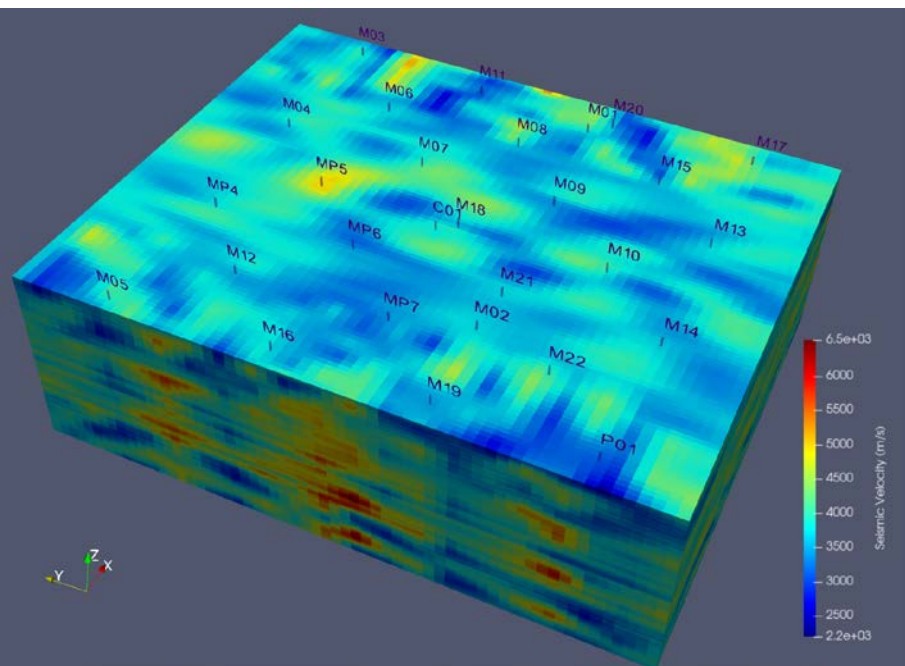

**Figure 6.** Three-dimensional seismic velocity model of the HES aquifer from 35 to 130 m below the ground surface.

The graph structure applied in pathfinding was acquired with i) the 3D seismic velocity model and ii) the borehole vertical

flow velocity profiles measured prior to the tracer experiments. The Boost Graph Library (Siek et al., 2001) was adopted to generate a bidirectional graph whose vertices were mapped onto the centers of subseismic model grid cells. The vertices were connected with the 26-neighborhood rule. Each pair of neighboring vertices was connected by two edges, one in each direction. This option is primarily useful for the implementation of borehole flow patterns, as discussed below. In regard to all vertex pairs located between the boreholes and/or corresponding to borehole sections without major vertical flow, the two connecting

edges were assigned identical weights. The weights were calculated as the arithmetic mean of the seismic velocity values multiplied by the Euclidean distance between the two vertices. This second factor corrected for the nonuniformity of the seismic model grid in all three spatial directions. In regard to the other edges, i.e., those corresponding to borehole sections where vertical flows were identified, the weights were calculated in two steps. First, the method described above was adopted, and a correction factor was then applied to promote any edges oriented along the flow direction, and conversely penalize edges



in the opposite direction. Since the vertical flow patterns and velocities in boreholes are basically dependent on the active pumping well, different edge-weight corrections were applied to the graph prior to searching for the optimal paths to wells M06, M07, M22 and MP6. The overall velocity range based on the borehole flowmeter surveys was $0 - 3.5$ m/min. Edge-weight corrections were applied to any borehole sections with a flow velocity higher than 0.1 m/min. The (multiplication/division) correction factor was empirically set to 15 times the velocity in m/min. This correction was also

applied to the pumped well considering a vertical flow velocity calculated based on the pumping flow rate divided by the borehole cross-section. The implemented graph structure comprised 1094400 vertices and 15039782 edges.

The conduit network obtained with the ARLib OnePass+ algorithm (please refer to section 3) is shown Fig. 7. As is the rule when modeling real groundwater systems with contrast to synthetic studies, the obtained results cannot be validated but can only be (eventually) invalidated (Konikow and Bredehoeft, 1992). More specifically, strict validation would require new

boreholes to be drilled to verify if they actually intersect the identified conduits. This operation is not envisaged at the HES, both because of its financial cost and because the creation of new boreholes could modify the flow-path structure by creating new bypasses between the different karst horizons.

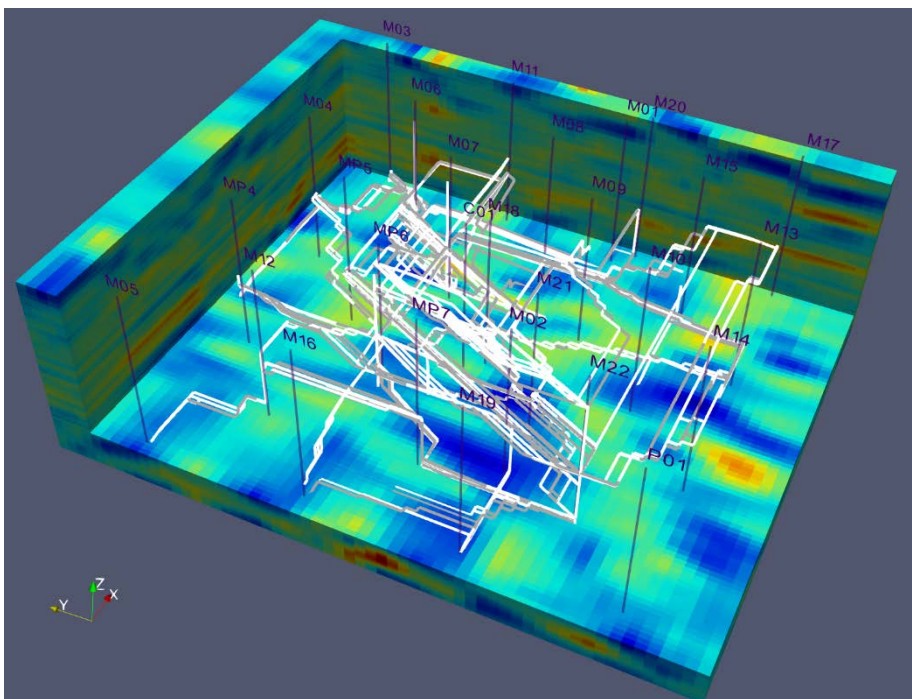

**Figure 7.** HES karst conduit network obtained from the combined analysis of interwell tracer tests, borehole flowmeter logs

and 3D seismic imaging data. The white and grey lines represent the main and secondary paths, respectively. The main paths are those identified via multiflow 2RNE inversion of the tracer BTCs and the secondary paths are the additional paths identified via multiflow ADE inversion; please refer to Table A1.





Of the independent data against which the results can be confronted, it can be noted that i) the computed network shows a layered structure that is consistent with the known karst horizons at 50, 90, and 115 m depth, and ii) the statistical directional

analysis of the conduits is also consistent with the local structural directions (Fig. 8).

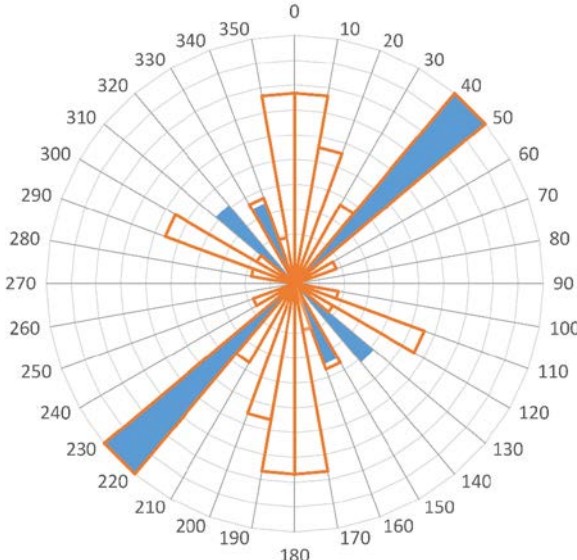

**Figure 8.** Normalized rose diagram of computed conduits (solid sectors) and measured fractures on a rock outcrop located 4 km west of the HES (empty sectors). The N130-140 direction is not locally expressed on this outcrop, but it is a well known regional karst direction, e.g., Bodin and Razack (1997). Conversely, the N-S and N110-120 fracture directions not found in

the karst conduit inversion should not be considered of concern as it is well known that karst exploits only part of the original rock discontinuities (Bodin and Razack 1997; Häuselmann et al., 1999).

A very partial but direct verification of the computed network is also possible based on its effective accessible parts, i.e., where the paths follow the vertical portions of boreholes. In the network, three boreholes likely play a vertical relay role between the different subhorizontal karst horizons: borehole MP7 between well M12 and pumped well M22, borehole M12 between M05

and MP6, and borehole M02 between M19 and MP6. As a pump was still present in MP6 when the network computation was completed, tracing experiments from M05 and M19 were again performed but with complementary concentration monitoring in intermediate boreholes M12 and M02 by means of fluorometers installed at depth. Other fluorometers were also installed at depth in nearby boreholes not expected to be in the tracer path. In terms of tracer experiment M05-MP6, the concentrations were monitored in MP4, M16, MP6 and M12. In terms of tracer experiment M19-MP6, boreholes MP7, M16, and M21 were

monitored in addition to boreholes MP6 and M02. The obtained results, as shown in Fig. 9, are mostly as expected since the tracer actually flowed through intermediate boreholes M12 and M02 and was not detected in boreholes M16, MP4, and MP7. The only surprise was the observed unforeseen vertical transfer of the tracer through M21. These experimental results support





the overall reliability of the calculated network but also expose its incompleteness. To test whether this incompleteness could occur due to the settings of the ARLib OnePass+ algorithm, the paths between M19 and MP6 were recalculated by lowering

the value of parameter $\theta$ (Eq. 1) from 0.5 to 0.05 to increase interpath diversification. No path passing through M21 was identified. It is therefore likely that the incompleteness (or partial incorrectness) of the conduit network is primarily due to the resolution limits of the seismic data.

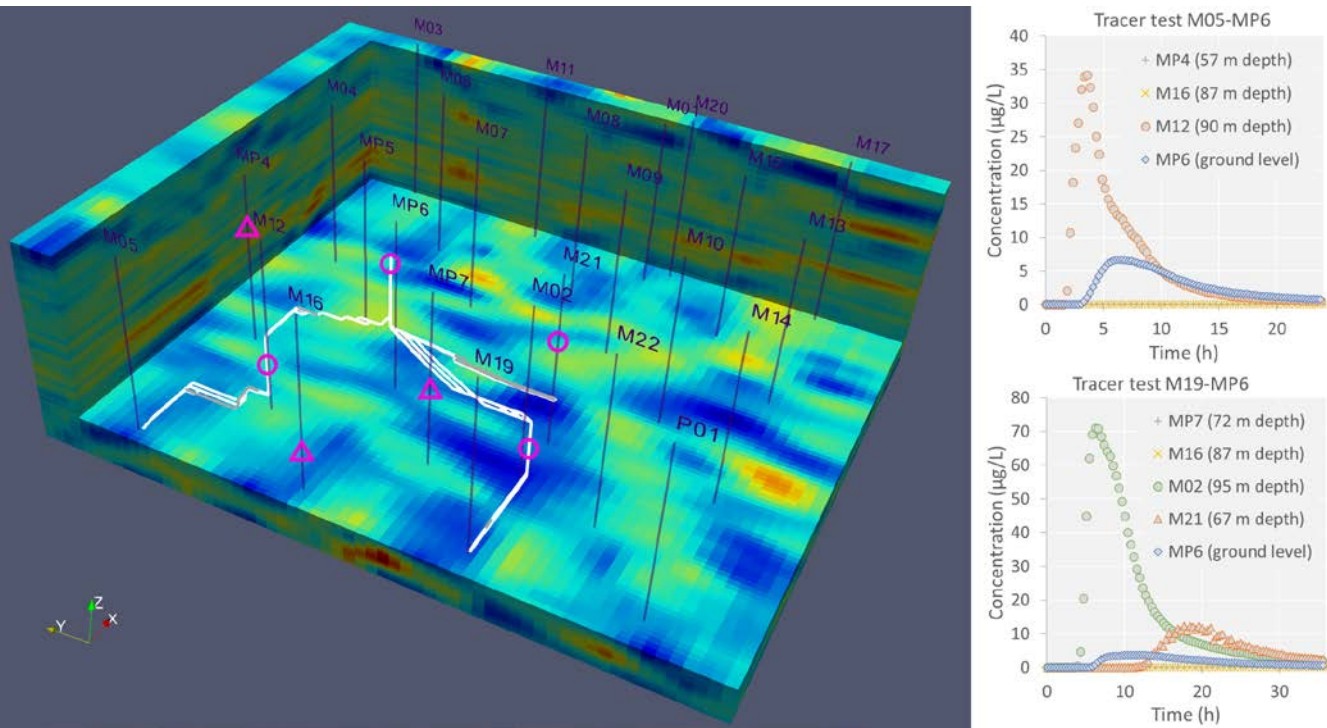

**Figure 9.** Verification tracer tests conducted between wells M05 (tracer injected at a 111 m depth), M19 (tracer injected at a
111 m depth) and MP6 (pumping well). The circles and triangles indicate the location of the fluorometer probes (Albillia GGUN FL22) that were or were not in the tracer path respectively. The paths shown are those that have potentially been used by the tracer. The results obtained thus partially validate the paths M05-M12-MP6 and M19-M02-MP6. The main unknown is the connection between M19 and M21, especially as the vertical flow in M21 is downward.

## 5 Discussion and conclusions

The approach outlined in this paper aims to improve the assessment of conduit network geometry in karst aquifers. The prerequisite data for the application of the method are i) a set of BTCs corresponding to tracer experiments conducted between different points in the karst network whose geometry is to be specified, and ii) a 3D model of subsurface heterogeneity derived from geophysical methods sensitive to karst features. As discussed in the review of Chalikakis et al. (2011), most geophysical





investigation methods are to a certain extent sensitive to karst features because the physical properties of a karst conduit filled
with water are generally different from those of the surrounding rock. However, the downside is that the amplitude of the
anomalies associated with karst conduits is often small, which results in an insufficient resolving capacity of geophysical
methods to capture the geometry of karst networks unambiguously and accurately. The main strength of the approach we
propose is to supplement the information content of geophysical surveys with the results of tracer BTC analysis. Part of the
ambiguity in the identification of karst conduits based on geophysical data can be alleviated by building on the prior estimation
of the number of conduits between pairs of point locations. The first step of the method is therefore to analyze each BTC using
a regularized multiflow inversion method, as implemented in the MFIT software, to determine the minimum number of distinct
paths used by the tracer between injection and monitoring points. The second step consists of implementing a graph based on
the 3D geophysical model of subsurface heterogeneity and calculating the previously enumerated paths via the application of
an alternative SP algorithm. The source code of the program used in this study is accessible via a DOI link provided below in
the "Code availability" section below and can be adapted by the interested user.

With the use of HES tracer test data, 3D seismic data, and borehole flowmeter logs to illustrate our approach, we demonstrated
the feasibility of delineating discrete karst conduit networks. The reliability of the identified network was locally verified
through new tracer experiments performed after completion of the computations. To our knowledge, the present paper is the
first to combine tracer and geophysical data to identify the discrete geometry of a karst network. Another conceivable approach
would be the joint inversion of both types of data, but inversion methods based on a discrete approach to flow and/or transport
paths are still at an early stage of development, e.g., Somogyvári et al., 2017 and Fischer et al., 2020.

The main limitations of the proposed approach are i) the spatial density of the tracer tests required to constrain the number of
paths between the nodes of the network, and ii) the sensitivity and/or spatial resolution problems inherent to geophysical
methods. The M19-M21-MP6 pathway not identified from the HES data but highlighted a posteriori during the verification
tracer tests is a concrete illustration of the limitations of the method. It must also be acknowledged that this approach allows
the determination of only the backbone structure of the network and not the complete geometry of the karst conduit system,
including the cross-sectional dimension and shape of conduits. Nevertheless, the implemented approach achieves substantial
progress toward the characterization and subsequent modeling of karst aquifers. To complement the transport parameters of
the individual karst conduits already inverted with MFIT, the computed network may be readily incorporated into hybrid
discrete conduit-continuum models such as MODFLOW-USG (Panday et al., 2013) to calibrate the hydraulic parameters of
the determined karst conduits and surrounding rock matrix against pumping test data. Finally, the approach may also be
applicable to other types of aquifers in the delineation of preferential flow paths induced by subsurface heterogeneity provided
that geophysical methods are sensitive to this heterogeneity.





## Appendix A: Flow-path number analysis of HES tracer BTCs with MFIT software

350 **Table A1.** Minimum number N* of flow paths between the tracer injection and pumping well pairs identified with the multiflow ADE (MDMi) model and multiflow 2RNE (MDP-2RNE) model.

| Tracer experiment (injection well-pumping well) | $N^*$ MDMi | $N^*$ MDP-2RNE |
|---|---|---|
| M11-M06 | 3 | 2 |
| M04-M06 | 6 | 3 |
| M03-M06 | 3 | 2 |
| M20-M06 | 6 | 4 |
| MP5-M06 | 2 | 2 |
| M07-M06 | 6 | 4 |
| MP4-M06 | 4 | 3 |
| M06-M07 | 5 | 2 |
| M20-M07 | 5 | 3 |
| M13-M07 | 6 | 3 |
| M21-M07 | 4 | 3 |
| M15-M07 | 3 | 2 |
| MP6-M07 | 5 | 4 |
| MP5-M07 | 4 | 2 |
| M03-M07 | 7 | 4 |
| M11-M07 | 3 | 3 |
| MP7-M22 | 6 | 4 |
| M21-M22 | 6 | 3 |
| M19-M22 | 5 | 4 |
| M16-M22 | 5 | 3 |
| M13-M22 | 6 | 3 |
| M09-M22 | 2 | 1 |
| M06-M22 | 6 | 3 |
| M15-M22 | 3 | 2 |
| M20-M22 | 8 | 5 |
| M11-M22 | 4 | 2 |
| M12-M22 | 7 | 3 |
| MP6-M22 | 11 | 10 |
| MP5-M22 | 6 | 3 |
| MP4-M22 | 5 | 2 |
| M17-M22 | 5 | 4 |
| M21-MP6 | 6 | 2 |
| M07-MP6 | 6 | 3 |
| M12-MP6 | 4 | 2 |
| MP5-MP6 | 5 | 3 |
| MP4-MP6 | 4 | 3 |
| M16-MP6 | 5 | 4 |
| M05-MP6 | 4 | 3 |
| MP7-MP6 | 6 | 3 |



| | | |
|---|---|---|
| M02-MP6 | 5 | 3 |
| M11-MP6 | 4 | 2 |
| M17-MP6 | 4 | 2 |
| M13-MP6 | 5 | 3 |
| M20-MP6 | 6 | 4 |
| M19-MP6 | 7 | 6 |
| M22-MP6 | 5 | 3 |
| M06-MP6 | 6 | 3 |
| M04-MP6 | 5 | 2 |
| M15-MP6 | 4 | 2 |
| M03-MP6 | 5 | 2 |

**Code availability.** The MFIT program is available from https://doi.org/10.5281/zenodo.3470751 (Bodin, 2020) under the terms of the CeCILL Free Software License Agreement v2.1. The kPOP program used for graph generation and computation of the alternative paths between pairs of tracer injection and monitoring points was written in C++. The source code of the kPOP program is available from https://doi.org/10.5281/zenodo.4487305 (Bodin, 2021) under the terms of the CeCILL Free Software License Agreement v2.1.

**Data availability.** The HES tracer test data, seismic data, and borehole flowmeter logs processed in section 4 of this study are available from the H+ database (http://hplus.ore.fr/en/poitiers/data-poitiers, last accessed: 10 November 2021) with registration of a free account.

**Author contribution.** JB designed the research. GP and BN carried out the borehole flowmeter measurements and data analysis, and developed the experimental protocol used for the HES tracer experiments. All co-authors contributed to the field tracer experiments. BN pre-processed the borehole flowmeter data and the tracer test data and implemented their insertion into the H+ database. JB performed the inverse modelling of the tracer test data and implemented the code for the k-shortest path computation. All authors discussed the results. JB prepared the manuscript with contributions from all co-authors.

**Competing interests.** The authors declare that they have no conflict of interest.

**Financial support.** This research was supported by the French National Observatory H+, the European Union (ERDF), and "Région Nouvelle Aquitaine".

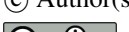


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
