# Peer review of "Delineation of discrete conduit networks in karst aquifers via combined analysis of tracer tests and geophysical data"

_Hydrology and Earth System Sciences, 2021_

## Referee Comment (RC1)

**Review of the manuscript hess-2021-584**
**"Delineation of discrete conduit networks in karst aquifers via combined analysis of tracer tests and geophysical data" by Bodin J, Porel G, Nauleau B, Pasquet D**

January 6, 2022

**1    General comments**

This manuscript is devoted to the development of a procedure based on the analysis of tracer tests and seismic data to map the conduit network of a karst aquifer.

The work is interesting. From the scientific point of view, the description of the BTC fitting procedure should be improved, because it is not thorough, sometimes is not clear and rigorous.

The manuscript is well organized and well written. However, some sentences could be misleading and some terminology is not fully appropriate.

I think that the overall quality is sufficient and the manuscript could be published, after a moderate to major revision.

**2    Specific comments**

1. Lines 38 & 39. The statement "the amplitude of the geophysical anomalies associated with karst conduits highly depends on their size and depth" is quite generic. The same sentence could be used for the application of geophysical methods to any geological environment, e.g., one could write that "the amplitude of the geophysical anomalies associated with sedimentary structures in alluvial aquifers highly depends on their size (thickness and lateral extension) and depth". Specific problems related with the application of geophysical methods to map karst structures (a mixture of "linear" conduits and "volumetric" cavities) is given in a more appropriate way at lines 319ff.

2. Line 44. I would avoid sentences like "it has been increasingly agreed that geophysical imaging methods are not silver bullets". This is an example of generic statements which do not provide any scientific, rigorous message.

3. Lines 48 & 49. May be, I do not properly understand the sentence "To the authors' knowledge, the only previous study to map karst conduit networks based on geophysical data is that of Vuilleumier et al. (2013)", but it sounds strange to me. In fact, the manuscript lists several papers which deal with the problem of mapping conduit networks in karst areas with geophysical methods. Moreover, Chalikakis et al. (2011) is cited as a review paper on this topic. A fast check with on-line search engines, like Google Scholar, provides some other papers which show the use of geophysical data to map karst features (e.g., `DOI:10.1016/j.crte.2009.08.005` and `DOI:10.1007/s10064-004-0247-4`).

4. There is some confusion in the definition and application of the inverse problem.

   (a) Lines 108. Which weight is used in the definition of $PHI$?

   (b) Lines 110, 11 & 124. The work by Vilfredo Pareto is associated to multi-objective optimization. This paper does not consider such an approach. Moreover, "Pareto curve" is used either to indicate the Pareto set, when dealing with two objective functions, or to denote cumulative curves in statistical software packages.

   (c) Lines 117 to 120. Regularization is obviously useful to design stable methods of solution, but it would be important to give more details about the parameters used and to discuss the sensitivity of fitted values with respect to such parameters. Also it would be important to discuss the characteristics of $PHI$ (local minima, flat behavior around the minimum, etc.).

   (d) The choice of $N^*$ is somehow arbitrary. The criteria given in the text are not strict. From the plots in the upper part of Figure 2, I would consider acceptable $N^* = 2$ for MFIT-2RNE, because adding another channel does not significantly improve the fit. I think it would be necessary to consider error measurements

to choose the value of $N^*$. For instance, one could fix a threshold for $PHI$, say $PHI^*$, physically congruent with the expected uncertainty in the data, so that $N^*$ could be chosen as the smallest value of $N$ such that $PHI(N) < PHI^*$.

(e) Line 120, figure 2. Is $PHI$ a dimensionless quantity?

(f) Line 123. The squared errors are weighted (see specific comment #4a above), aren't they?

(g) Line 335. What is meant by "inversion methods based on a discrete approach to flow and/or transport paths"? Details should be given.

5. Some details about the data are missing or not precise.

(a) Information about diameter, casing and screened intervals of the boreholes are missing.

(b) The depth below the ground surface of the three lithostratigraphic units where karst feature are dominant is not given in a precise way in this manuscript and in the referenced papers. I list the data in Table 1. There is a lot of confusion, which must be fixed.

(c) The papers to which the reader is referred for details about the seismic reflection survey are rather disappointing, as they are affected by a certain degree of self-plagiarism, in the sense that some paragraphs and figures are "copy-and-pasted" among the papers. In my opinion, the most interesting for the content of this manuscript is Mari and Porel (2008). My suggestion is to accompany the citation in the text with a short description of the differences between the four cited papers.

(d) The processing sequence for reflection seismic data does not include migration. Are the reflectors poorly inclined?

| Paper | shallow unit | intermediate unit | deep unit |
|---|---|---|---|
| This manuscript | 50 | 90 | 115 |
| Mari and Porel (2008) | 50 | 88 | 115 |
| Mari et al. (2009) | 35 to 40 | 85 to 87 | 110 to 115 |
| Mari and Porel (2018) | 35 | 88 | 110 |
| Mari et al. (2020) | 35 | 88 | 110 |

Table 1: Depths (in meters b.g.l.) of the lithostratigraphic units hosting karst features.

**3   Technical comments**

1. Line 43. Expression "applicable to very small and deep karst conduits" is misleading and should be rephrased.

2. Line 49. "This" should be substituted with "that".

3. Line 69. I have never seen the use of the expression "tracer-labeled solution". "Tracer" is enough.

4. Lines 71 & 72. Remark "with a much shorter duration of the injection signal than the mean tracer transit time" could be expanded and specified. Is this remark given to support the approximation of a pulse injection? If so, it should be explicitly stated.

5. Line 80. "Heavy" should be substituted with "long".

6. Lines 132 & 133. Expression "geophysical field" is not precise. The gravity field, the magnetic field, the geothermal field are "geophysical fields".

7. Equation (1). The explanation of this equation is rather cumbersome, but at the very end the concept is rather simple. I think it can be described in a better way. The symbol "$\forall$" can be erased.

8. Line 174. Expression "as shown in Fig. 1" is misleading. In fact it shows disconnected paths, whereas this sentence talks about "paths... not supposed to be fully disconnected".

9. Lines 245 & 475. "2021" should be corrected with "2018".

10. Line 248. Add the years of publication after "Mari et al.".

11. Line 265. Is "opposite" the right word? May be, "perpendicular"?

12. Line 267. "The overall velocity range based on the borehole flowmeter surveys was 0 - 3.5 m/min" should be substituted, possibly as "The maximum velocity measured during the borehole flowmeter surveys was 3.5 m/min".

13. Line 272. Add "in" after "is shown".

14. Line 273. Expression "with contrast to synthetic studies" can be erased.

15. Lines 288 to 291. The sentences "The N130-140 direction... Häuselmann et al., 1999)" should be moved in the text.

16. Line 291. Häuselmann et al. (1999) is missing in the reference list, isn't it?

17. Line 292. The sentence should be rephrased, because, in the present version, "its" refers to "computed network", whereas I assume it refers to the real karst network.

18. Lines 323 & 330. The sentence "The source code... the interested user" can be erased, as it merely repeats what is clearly and extensively written in the specific section.

19. Appendix A. I think that the table is not relevant. It would be much more informative a figure with two histograms or box plots showing the distribution of $N^*$ for the two models. Also, it would be interesting to see the trend of $PHI(N)$ for each case.

20. Line 405. Add "`DOI:10.5441/002/edbt.2017.37`".

21. Line 441. "Springer" should be corrected.

22. Line 490. "Springer" should be added before "Netherlands".

23. Lines 502 & 503. "In 2015 International Conference on Industrial Engineering and Systems Management (IESM)" should be added before "edited". Number of pages (663–669) and DOI (`10.1109/IESM.2015.7380229`) should be added.

---

## Author Response (AR1)

**Response to comments from Referee 1**

We sincerely thank Anonymous Referee #1 (AR1) for his or her thorough and thoughtful assessment of our manuscript. We respond below to the individual points.

Font legend:

1. *Referee's comments are shown in black italic font.*
2. Our responses are shown in a blue normal font.
3. The text that has been modified or added to the revised manuscript is shown in an orange normal font.

*AR1: This manuscript is devoted to the development of a procedure based on the analysis of tracer tests and seismic data to map the conduit network of a karst aquifer. The work is interesting. From the scientific point of view, the description of the BTC fitting procedure should be improved, because it is not thorough, sometimes is not clear and rigorous. The manuscript is well organized and well written. However, some sentences could be misleading and some terminology is not fully appropriate. I think that the overall quality is sufficient and the manuscript could be published, after a moderate to major revision.*

**RESPONSE:** We thank the referee for his or her positive comments. We recognize that some technical specifics of the BTC inversion procedure were not sufficiently discussed. This issue, as well as all other comments, has been addressed in the revised manuscript in the manner we detail below.

*AR1, Specific Comment #1 (SC1): Lines 38 & 39. The statement "the amplitude of the geophysical anomalies associated with karst conduits highly depends on their size and depth" is quite generic. The same sentence could be used for the application of geophysical methods to any geological environment, e.g., one could write that "the amplitude of the geophysical anomalies associated with sedimentary structures in alluvial aquifers highly depends on their size (thickness and lateral extension) and depth". Specific problems related with the application of geophysical methods to map karst structures (a mixture of "linear" conduits and "volumetric" cavities) is given in a more appropriate way at lines 319ff.*

**RESPONSE:** We amended this section as follows:

However, the detection and mapping of karst features (conduits and/or cavities) based on geophysical surveys remains a challenging task due to their volumetrically small proportion in rock volumes that may be intrinsically associated with other types of spatial heterogeneity, e.g., sedimentary facies variations. Bechtel et al. (2007) and Chalikakis et al. (2011) reviewed the strengths and weaknesses of different geophysical methods that can be considered for locating karst features in the subsurface. To date, reported field applications of surface geophysical methods to locate known or suspected water-filled karst conduits have only been successful (…)

The following reference has been added to the revised manuscript: Bechtel, T. D., Bosch, F. P., and Gurk, M.: Geophysical methods, in: Methods in Karst Hydrogeology, Taylor & Francis/Balkema, Leiden, The Netherlands, 171–199, 2007.

*AR1, SC2: Line 44. I would avoid sentences like "it has been increasingly agreed that geophysical imaging methods are not silver bullets". This is an example of generic statements which do not provide any scientific, rigorous message.*

**RESPONSE:** In our original manuscript, "silver bullets" were in quotation marks because they were part of the title of a guest editorial paper by K. Singha in the journal Groundwater. Nevertheless, since this part of the sentence and the associated reference were not essential, we removed both in the revised manuscript.

*AR1, SC3: Lines 48 & 49. May be, I do not properly understand the sentence "To the authors' knowledge, the only previous study to map karst conduit networks based on geophysical data is that of Vuilleumier et al. (2013)", but it sounds strange to me. In fact, the manuscript lists several papers which*

*deal with the problem of mapping conduit networks in karst areas with geophysical methods. Moreover, Chalikakis et al. (2011) is cited as a review paper on this topic. A fast check with on-line search engines, like Google Scholar, provides some other papers which show the use of geophysical data to map karst features (e.g., DOI:10.1016/j.crte.2009.08.005 and DOI:10.1007/s10064-004-0247-4).*

**RESPONSE:** In our manuscript, we distinctly use the terms "karst conduit" (or "karst feature") and "karst conduit network" (or "karst network"), as there is a fundamental difference in size and geometric complexity between the two. A karst network is a collection of karst conduits, and its overall geometric structure can be very complex. The article by Vuilleumier et al. (2013) is actually the only one that, to our knowledge, addresses the identification of a realistic/complex karst network from geophysical data. In the other papers cited in our manuscript, as well as in the additional references provided by the reviewer, the scale of investigation is that of a single conduit or that of extremely simple networks consisting of at most two or three conduits; see, for example, the abstract in DOI:10.1016/j.crte.2009.08.005 and Fig. 5 in DOI:10.1007/s10064-004-0247-4.

***AR1, SC4a:*** *There is some confusion in the definition and application of the inverse problem. Lines 108. Which weight is used in the definition of PHI?*

**RESPONSE:** In MFIT, the user has the possibility to weight some observations more than others if he or she wants the minimization of PHI to result in a preferential fitting of the model to these observations. However, in the present study, we used an identical weight (of 1) for each observation. We removed the term "weighted" in the definition of PHI.

***AR1, SC4b:*** *Lines 110, 11 & 124. The work by Vilfredo Pareto is associated to multi-objective optimization. This paper does not consider such an approach. Moreover, "Pareto curve" is used either to indicate the Pareto set, when dealing with two objective functions, or to denote cumulative curves in statistical software packages.*

**RESPONSE:** The identification of $N^*$ requires minimizing both PHI($N$) and $N$, hence a notion of multiobjective optimization and our reference to Pareto's work. However, we acknowledge that this reference is not fully rigorous since the minimization of PHI($N$) and $N$ is not simultaneous. The automatic optimizations carried out using PEST are only interested in minimizing PHI for a given $N$, the minimization of $N$ being performed in a second step by graphical analysis of the PHI($N$) curve. We removed the words "Parto curve" in the revised manuscript.

***AR1, SC4c:*** *Lines 117 to 120. Regularization is obviously useful to design stable methods of solution, but it would be important to give more details about the parameters used and to discuss the sensitivity of fitted values with respect to such parameters. Also it would be important to discuss the characteristics of PHI (local minima, flat behavior around the minimum, etc.).*

**RESPONSE:** It is true that we did not give much detail in the original manuscript about the technical details of the BTC inversion procedure. The reason is twofold. First, these details are given in the article dedicated to the MFIT software (Bodin 2020), and second, because as soon as such discussion is open it is difficult to make it simple and short because it is then necessary to evoke a set of elaborated concepts and methods. We have expanded the discussion of these issues in the revised manuscript, trying to be as synthetic as possible so that the reader does not lose the thread, leading to the identification of the minimum number of distinct paths used by the tracer between the injection and monitoring points, as follows:

The fitting process of a model curve to an experimental tracer BTC first requires specification of the number of channels $N$, followed by optimization of the model parameters pertaining to each channel. The main features of the PEST optimization algorithm are summarized below. We refer interested readers to (Doherty and Hunt, 2010) and (Doherty, 2015) for a more comprehensive presentation of theoretical concepts and associated methods and to Bodin (2020) for their specific implementation in MFIT software dedicated to tracer BTC fitting.

The PEST optimization routines are primarily based on the Gauss Marquardt Levenberg Algorithm (GMLA). The objective function that is minimized during the optimization process is defined as the sum of two terms. The first term is the "measurement objective function" PHI, which is defined as the sum of the squared differences between the tracer BTC and the model-simulated curve. The second term is referred to as the "Tikhonov regularization objective function" and acts as a penalty function for deviations from some preferred parameter conditions. In the present study, we used regularization constraints that promote a solution of minimum variance for the model parameters pertaining to the different channels. The Tikhonov regularization contributes to the stability of the numerical optimization scheme, jointly with the singular value decomposition (SVD) method that removes from the estimation process the combinations of model parameters for which the tracer BTC is uninformative. Tikhonov regularization also allows us to prevent any overfitting of the tracer BTC. The regularization is controlled by a PEST variable called PHIMLIM, which defines a threshold for the objective function below which we consider that the model is calibrated. The PHIMLIM value should be congruent with both the uncertainty in the measured concentrations and the structural noise resulting from the inability of the models to perfectly simulate real-world processes. Mainly because of this last feature, it is generally not obvious to estimate a priori what is a suitable PHIMLIM value. In the present study, we used a strategy suggested by Doherty and Hunt (2010), which consists of setting PHIMLIM to a value slightly higher than the minimum objective function that can be achieved without applying regularization constraints. We chose to set PHIMLIM 15% above the minimum value of PHI that could be obtained using 15 channels. The main cost of this method is having to perform at least two optimization runs each time: first without and then with regularization. In fact, the actual number of optimization runs is much higher because MFIT also includes a "multistart" procedure that consists of repeating the optimization process starting from different initial parameter value sets. This procedure is intended to improve the chances of converging to the global minimum of the objective function rather than a local minimum, which is a well-known potential issue with the GMLA.

The determination of the minimum number of distinct tracer flow paths between injection and monitoring points, hereafter referred to as $N^*$, is achieved through analysis of the curve representing the minimum PHI value obtained at different $N$ values. The typical shape of a PHI($N$) curve is that of a monotonic decreasing function converging to a horizontal asymptote, which corresponds to the PHIMLIM threshold (Fig. 2). The $N^*$ value corresponds to the smallest value of $N$ such that PHI($N$) ≈ PHIMLIM. The $N^*$ value is also a model-dependent parameter, as fewer channels are required to fit a heavy-tailed BTC with a double-porosity model (SFDM or 2RNE) rather than with the ADE instantaneous injection model.

The following reference has been added to the revised manuscript: Doherty, J. E. and Hunt, R. J.: Approaches to highly parameterized inversion: a guide to using PEST for groundwater-model calibration. U.S. Geological Survey Scientific Investigations Report 2010–5169, 59 p., 2010.

[Figure]

**Figure 2.** Example of BTC fitting analysis with the multiflow ADE and 2RNE models for different numbers of channels $N$. The experimental BTC corresponds to a tracer test performed in 2016 at the HES between wells M16 (injection) and M22 (pumping and observation). PHI is the fitting error objective function (sum of the squared errors between the tracer BTC and model-simulated curve) minimized with the regularization routines in PEST. The decreasing trend of the PHI($N$) curves indicates the improvement in model fit with an increasing number of channels $N$. PHIMLIM is a threshold for PHI that prevents overfitting of the tracer BTC. The optimal numbers of channels determined with the ADE and 2RNE models are $N^* = 5$ and $N^* = 3$, respectively.

*AR1, SC4d1: The choice of N\* is somehow arbitrary. The criteria given in the text are not strict. From the plots inthe upper part of Figure 2, I would consider acceptable N\* = 2 for MFIT-2RNE, because adding anotherchannel does not significantly improve the fit.*

**RESPONSE:** The gain between $N^* = 2$ and $N^* = 3$ for the 2RNE model is difficult to visually assess in the upper part of Fig. 2 due to its small size, but the 3-channel model provides a better fit to the experimental points at and just after the concentration peak, where the experimental curve shows a slight inflection (see Fig. A below). This inflection provides information about the heterogeneity of the flow system that we feel is valuable to extract from the tracer test data.

[Figure]

Fig. A. Example of BTC fitting analysis with the multiflow 2RNE model for different numbers of channels (extracted from Fig. 2 of the original manuscript).

***AR1, SC4d2:*** *I think it would be necessary to consider error measurements to choose the value of N\*. For instance, one could fix a threshold for PHI, say PHI\*, physically congruent with the expected uncertainty in the data, so that N\* could be chosen as the smallest value of N such that PHI(N) < PHI\*.*

**RESPONSE:** We fully agree that the level of noise/uncertainty in the measured concentrations must be considered when identifying $N^*$. This is actually taken into account in our methodology, but we did not highlight this issue. This has been corrected in the revised manuscript as outlined in our response to comment SC4c.

***AR1, SC4e:*** *Line 120, figure 2. Is PHI a dimensionless quantity?*

**RESPONSE:** The unit of PHI ($mg^2\ l^{-2}$) has been added to Fig. 2.

***AR1, SC4f:*** *Line 123. The squared errors are weighted (see specific comment #4a above), aren't they?*

**RESPONSE:** This choice is left to the user of MFIT, but in this study, the same weight (= 1) was used for each difference. We removed the term "weighted" in the revised manuscript.

***AR1, SC4g:*** *Line 335. What is meant by "inversion methods based on a discrete approach to flow and/or transport paths"? Details should be given.*

**RESPONSE:** There are basically 3 main types of approaches for spatially distributed ("direct" or "inverse") numerical modeling of flow and mass transport in fractured and/or karst aquifers: 1) the continuous approach where fractures or karst conduits are not explicitly represented in the model and where the spatial heterogeneity of the aquifer is described via effective parameters associated with the grid cells of a continuous model, 2) the discrete approach where the model integrates an explicit representation of the fractures/karst conduits without a continuous grid, and 3) the hybrid approach combining the two approaches mentioned above. The point we raise in line L360 of our revised manuscript is that inverse modeling of tracer test data in fractured and/or karst aquifers based on approaches 2) and 3) and aimed in part at inverting/identifying the spatial distribution of discrete structures are to date very few, considering the published papers mentioned in the manuscript. We believe that readers of HESS who might be potentially interested in this manuscript are assumed to be familiar with the notion of continuous vs. discrete modeling, and so we do not believe it is necessary to discuss these concepts further.

***AR1, SC5a:*** *Information about diameter, casing and screened intervals of the boreholes are missing.*

**RESPONSE:** The reference (Nauleau et al., 2022), associated with a DOI (https://doi.org/10.26169/hplus.poitiers_technical_logs) indexed on the database of the French National

Observatory H+ and providing the technical specifications of the boreholes has been added in the general introduction of the HES (line 203 of the revised MS).

*AR1, SC5b: The depth below the ground surface of the three lithostratigraphic units where karst feature are dominant is not given in a precise way in this manuscript and in the referenced papers. I list the data in Table 1. There is a lot of confusion, which must be fixed.*

| Paper | shallow unit | intermediate unit | deep unit |
|---|---|---|---|
| This manuscript | 50 | 90 | 115 |
| Mari and Porel (2008) | 50 | 88 | 115 |
| Mari et al. (2009) | 35 to 40 | 85 to 87 | 110 to 115 |
| Mari and Porel (2018) | 35 | 88 | 110 |
| Mari et al. (2020) | 35 | 88 | 110 |

Table 1: Depths (in meters b.g.l.) of the lithostratigraphic units hosting karst features.

**RESPONSE:** As mentioned in line 205 of the revised manuscript, the lithostratigraphic units are nearly but not perfectly horizontal. It is therefore difficult to specify a single depth value for the karst horizons: a dip of 1.5° over a distance of 200 m (characteristic size of the HES) represents a difference in elevation of 5 m, hence the small differences between the depths of the intermediate and deep units in Table 1 of the referee. The larger discrepancies for the shallow unit can be explained by the fact that there are actually not 3 but 4 karst horizons. The karstic signature of the 35-40 m horizon is more marked in the seismic data than the 50 m horizon, which is why the latter is sometimes not mentioned in the articles by Mari et al. On the other hand, the 35-40 m horizon is not "visible" in the hydrogeological investigations (borehole flow surveys, borehole imaging, and interwell tracer tests) because most of the boreholes are equipped with solid steel or PVC casing at this depth. Since our method of identifying karst conduits is preconditioned by the analysis of borehole flow logs and interwell tracer test data, the karst features at 35-40 m depth are basically unidentifiable. We clarified this in the revised manuscript by adding the following text: "According to Mari et al. (2009), the seismic surveys suggest the existence of an additional karst horizon between 35 and 40 m depth, but since most of the boreholes are equipped with solid steel or PVC casing at this depth, no interwell tracer test data are associated with this horizon. Since our approach for the delineation of karst networks is preconditioned by the analysis of such tracer test data, the karst features at 35-40 m depth are basically unidentifiable and will not be addressed below."

*AR1, SC5c: The papers to which the reader is referred for details about the seismic reflection survey are rather disappointing, as they are affected by a certain degree of self-plagiarism, in the sense that some paragraphs and figures are "copy-and-pasted" among the papers. In my opinion, the most interesting for the content of this manuscript is Mari and Porel (2008). My suggestion is to accompany the citation in the text with a short description of the differences between the four cited papers.*

**RESPONSE:** We removed the references to the Mari and Porel (2018) and Mari et al. (2020) articles. The Mari and Porel (2008) article is now cited as the primary reference to the seismic data, and a citation to the Mari et al. (2009) paper has been retained to support the discussion of the unstudied shallow karst horizon at 35-40 m depth.

*AR1, SC5d: The processing sequence for reflection seismic data does not include migration. Are the reflectors poorly inclined?*

**RESPONSE:** Yes indeed. As stated in line 205 of the revised manuscript and in the article by Audouin et al. (2008), the lithostratigraphic units are subhorizontal, dipping less than 2°.

***AR1, Technical Comment #1 (TC1):*** *Line 43. Expression "applicable to very small and deep karst conduits" is misleading and should be rephrased.*

**RESPONSE:** We removed this part of the sentence in the revised manuscript.

***AR1, TC02-03:*** Has been revised accordingly.

***AR1, TC04:*** *Lines 71 & 72. Remark "with a much shorter duration of the injection signal than the mean tracer transit time" could be expanded and specified. Is this remark given to support the approximation of a pulse injection? If so, it should be explicitly stated.*

**RESPONSE:** We added the following sentence to the revised manuscript: "This last assumption supports the approximation of a pulse injection as a boundary condition in the analytical transport models used later in this work."

***AR1, TC05:*** Has been revised accordingly.

***AR1, TC06:*** *Lines 132 & 133. Expression "geophysical field" is not precise. The gravity field, the magnetic field, the geothermal field are "geophysical fields".*

**RESPONSE:** We changed "geophysical field" to "geophysical survey data". This expression remains intentionally broad because at this point of the manuscript, we are still in the general introduction of the concept of our approach, and even if in our application example we use seismic data, our approach could just as well be applied to other types of geophysical measurements (e.g., electrical, electromagnetic, microgravity, etc.), provided that the geophysical method is sensitive to hydraulic conductivity variations.

***AR1, TC07:*** *Equation (1). The explanation of this equation is rather cumbersome, but at the very end, the concept is rather simple. I think it can be described in a better way. The symbol "$\forall$" can be erased.*

**RESPONSE:** We replaced Equation (1) and the associated text with three sentences as follows: "The main interest of this algorithm is that it allows the user to specify a maximum overlap ratio between alternative paths. During the search process, a candidate alternative path is successively compared to the previously retained paths by computing the ratio between the sum of the weights of the shared edges and the total cost of each of the previously retained paths, omitting the starting and ending edges. The candidate path is added to the KSP solution set only if its overlap ratio is below a predefined $\theta$ threshold value between 0 and 1."

***AR1, TC08:*** *Line 174. Expression "as shown in Fig. 1" is misleading. In fact, it shows disconnected paths, whereas this sentence talks about "paths... not supposed to be fully disconnected".*

**RESPONSE:** We changed the sentence to "As already noted, the $N*$ paths yielded by the inversion of tracer test data are not supposed to be as disconnected as in Fig. 1, but their overlap must nevertheless be small enough to produce a detectable signature in the BTC."

***AR1, TC09:*** *"2021" should be corrected with "2018".*

**RESPONSE:** As discussed in our response to comment SC5c, this reference has been removed.

***AR1, TC10:*** Has been revised accordingly

***AR1, TC11:*** *Line 265. Is "opposite" the right word? May be, "perpendicular"?*

**RESPONSE:** We changed "opposite" to "reverse". Actually, the type of graph used is birectional, which means that an edge connecting two nodes A and B has two different weights depending on the direction A to B or B to A.

***AR1, TC12-15:*** Has been revised accordingly

***AR1, TC16:*** *Line 291. Häuselmann et al. (1999) is missing in the reference list, isn't it?*

**RESPONSE:** Yes, it is. Thank you for your vigilance. The reference "Häuselmann, P., Jeannin, P.-Y., and Bitterli, T.: Relationships between karst and tectonics: case-study of the cave system north of Lake Thun (Bern, Switzerland), Geodin. Acta, 12, 377–387, https://doi.org/10.1080/09853111.1999.11105357, 1999" has been added to the revised manuscript.

*AR1, TC17: Line 292. The sentence should be rephrased, because, in the present version, "its" refers to "computed network", whereas I assume it refers to the real karst network*

**RESPONSE:** We rewrote the sentence as follows: "A very partial but direct verification of the computed network is also possible where the paths follow the vertical portions of boreholes."

*AR1, TC18:* Has been revised accordingly

*AR1, TC19: Appendix A. I think that the table is not relevant. It would be much more informative a figure with two histograms or box plots showing the distribution of N\* for the two models. Also, it would be interesting to see the trend of PHI(N) for each case.*

**RESPONSE:** Agree. The table has been replaced by a figure (Fig. 6) in the revised manuscript.

*AR1, TC20-23:* Has been revised accordingly
* * *
**Response to comments from Referee 2**

*This manuscript is a revised version of a manuscript I previously reviewed for another journal. My overall assessment of the previous version was minor revision and I also provided several comments. These comments have been addressed to my satisfaction in this version. This version also makes some changes to Introduction and Discussion, making it clear of the overall objective and limitations of their method. Overall, I think the manuscript is an improvement from the previous version and therefore suggest for publication. Below is my review for the previous version with minor comments removed.*

*This manuscript presents a novel approach to estimate discrete conduit network using seismic, vertical flow, and dye tracing data. The authors treat the question of estimating a subsurface conduit network as a k-shortest path (KSP) problem and nicely fit the three types of data to build a KSP model. Even though the estimated conduit network showed visible artifacts as admitted by the authors, the overall research approach is quite interesting. I particularly like how the authors combine three very different types of data into a single model. These artifacts may be just a reflection that vertical flow and dye tracing data were collected through boreholes/wells and some other parts of the aquifer have not been explored by the data collection methods. I recommend this manuscript to be accepted with minor revision. I just need some clarification regarding model result verification and explanation of primary/secondary paths. Here are my comments:*

> *1. Using two independent dye tracing data to verify the estimated conduit network is nice (Figure 7). But I can't see clearly which estimated paths are confirmed or missed by the dye tracing data (Figure 6). Instead of coloring the paths in figure 6, maybe a separate sub-image showing the estimated paths just for the area affected by each dye tracing can be added to Figure 7. Mark confirmed/missed path using different colors.*
> *2. I can't distinguish between primary and secondary paths from Figure 6. Maybe use different colors.*

We are very grateful to Junfeng Zhu for twice agreeing to review our work. His fair and constructive comments on a previous version allowed us to make substantial improvements and in particular led to Figures 8 and 10 of the revised manuscript, which are much more readable and informative than in the first manuscript. We are pleased that the reviewer is satisfied with our revisions, and we thank him again.